# Interactions between Chemesthesis and Taste: Role of TRPA1 and TRPV1

**DOI:** 10.3390/ijms22073360

**Published:** 2021-03-25

**Authors:** Mee-Ra Rhyu, Yiseul Kim, Vijay Lyall

**Affiliations:** 1Korea Food Research Institute, Wanju-gun 55365, Korea; kimys@kfri.re.kr; 2Department of Physiology and Biophysics, Virginia Commonwealth University, Richmond, VA 23298, USA; vijay.lyall@vcuhealth.org

**Keywords:** TRPV1, TRPA1, chemesthesis, Korean indigenous plants, taste, interaction

## Abstract

In addition to the sense of taste and olfaction, chemesthesis, the sensation of irritation, pungency, cooling, warmth, or burning elicited by spices and herbs, plays a central role in food consumption. Many plant-derived molecules demonstrate their chemesthetic properties via the opening of transient receptor potential ankyrin 1 (TRPA1) and transient receptor potential vanilloid 1 (TRPV1) channels. TRPA1 and TRPV1 are structurally related thermosensitive cation channels and are often co-expressed in sensory nerve endings. TRPA1 and TRPV1 can also indirectly influence some, but not all, primary taste qualities via the release of substance P and calcitonin gene-related peptide (CGRP) from trigeminal neurons and their subsequent effects on CGRP receptor expressed in Type III taste receptor cells. Here, we will review the effect of some chemesthetic agonists of TRPA1 and TRPV1 and their influence on bitter, sour, and salt taste qualities.

## 1. Introduction

In many cuisines, different parts of a plant including leaves, roots, bulbs, fruits, seeds, berries, kernels, and bark contain compounds that elicit unique flavors, and thus, are used to enhance the hedonic value of food. Plants utilize these flavoring compounds to defend themselves from external life-threatening organisms that cause disease and from predators, such as insects and vertebrates [1]. Humans from antiquity have used herbs (leafy part of the plants) and spices (from non-leafy parts of the plants) primarily for medicinal purposes and as food preservatives [2]. However, there is also is a long history of humans using herbs and spices for flavoring food and beverages to enhance the hedonic value of food. Historically, herbs and spices for flavoring food and beverages were introduced from Asian, Persian, Indian, and Greek cultures and following Alexander the Great’s campaigns in Central Asia around 330 B.C., and were disseminated to many regions and adopted by many cultures [3]. 

Chemosensation refers to a biochemical and sensory process of recognition of chemical molecules, by specialized transduction pathways in addition to those of smell and taste. Consumption of spices and herbs elicits the perception of irritation, pungency, cooling, warmth, or burning in the mouth. These sensations induced by spices and herbs are generally not considered to directly interact with the gustatory system but contribute to the chemical sense referred to as chemesthesis [4]. These sensations not only add to the hedonic aspect of food but also provide additional health benefits [5]. Chemicals that elicit chemesthesis activate the transient receptor potential (TRP) family of cation channels expressed in sensory nerve endings innervating the skin or mucosa in epithelial cells. A number of excellent reviews have discussed the role of TRP channels in chemesthesis, taste and pain [5,6,7,8,9,10,11,12].

TRP channels were initially described in *Drosophila.* Since then, TRP channels have been shown to be expressed in animals, choanoflagellates, apusozoans, alveolates, green algae, a variety of eukaryotes, but do not seem to be present in either Archaea or Bacteria. It is suggested that, over time, the primary role of TRP channels has evolved in signal transduction [13]. Many of the chemosensory ion channels are members of the thermosensitive TRP family including TRP ankyrin member 1 (TRPA1), TRP vanilloid member 1 (TRPV1), member 3 (TRPV3), member 4 (TRPV4), and TRP melastatin member 8 (TRPM8). TRPA1 was originally reported to respond to intense cold [14] and to oils of mustard and cinnamon, and many other natural chemicals that cause a burning sensation in humans [15]. However, it was later revealed that TRPA1 is species-specific in its response to cold temperature. In this respect, it was demonstrated that rodent but not human TRPA1 (hTRPA1) is cold-sensitive in vitro [16,17]. However, in contrast to the above study, cold-evoked hTRPA1 responses have been demonstrated both in whole-cell expression assays [18,19,20] as well as for purified hTRPA1 [21]. Cysteine residues critical to allyl isothiocyanate (AITC) sensing may be conserved across species [13,22]. In contrast, TRPV1 responds to temperatures at or above the pain threshold of heat, as well as to capsaicin, the main pungent ingredient in hot chili peppers [23]. TRPM8 responds to menthol and cooling innocuous range [24,25]. Both TRPA3 and TRPV4 respond to innocuous warming and to a number of plant-derived monoterpenoids [26] and flavones [27], respectively. Among the TRP channels, TRPV1 and TRPA1 are often co-expressed in trigeminal ganglion neurons, and are the most relevant ion channels in chemesthesis. These channels can be activated by spices and herbs and are well studied in terms of the molecular mechanisms involving chemically induced opening of these channels. Interestingly, it has also been reported that several unique constituents of plants, consumed as vegetables, that unlike herbs or spices, are not pungent, elicit unique flavor by opening TRPA1 and/or TRPV1 channels. Non-pungent compounds, such as some specific hydroxyl fatty acids in royal jelly [28], and capsiate, the main ingredient of a non-pungent cultivar of red pepper [29,30], activate TRPA1 and/or TRPV1. In addition to natural compounds, *N-*geranyl cyclopropylcarboxamide, a synthetic compound, acting as a salt and umami taste enhancer [31], also stimulates both TRPA1 and TRPV1. *N-*geranyl cyclopropylcarboxamide is of particular significance, because it is implicated in enhancing neural responses to salt in rodents and in modulating salt taste responses in human subjects, suggesting a bridge between chemesthesis and salt taste [31,32]. Intriguingly, recent studies suggest that some chemicals that primarily elicit chemesthesis can directly or indirectly influence some, but not all, primary taste qualities: sweet, bitter, umami, sour, and salty, thus providing a potential linkage between chemesthesis and taste [33]. Here we will review the potential role of TRPA1 and TRPV1 in chemesthesis and their interactions with primary taste qualities in the gustatory system.

## 2. TRPA1 and TRPV1: Key Chemosensory Ion Channels in Chemesthesis

### 2.1. TRPA1

#### 2.1.1. TRPA1 Structure, Expression and Function

TRPA1 is a member of the TRP superfamily of ion channels. All TRP channels are composed of 4 subunits that span the plasma membrane 6 times with cytoplasmic COOH- and NH_2_-terminal domains. These transmembrane segments are designated S1 through S6 and the channel pore region lies between S5 and S6, and most often shows a high Ca^2+^ permeability [34] (Figure 1). TRPA1 has a unique and unusual structural feature of harboring 14 ankyrin repeats in the NH_2_-terminal domain and is thought to be pertinent to the regulation of channel activity by intracellular components [35]. TRPA1 functions as one of the primary sensors of environmental irritants and noxious substances including diverse chemical compounds in plants, many of which elicit a pungent sensation. TRPA1 activators can be classified into electrophilic and non-electrophilic substances. Many herbs- and spices-derived TRPA1 agonists such as allyl isothiocyanate (AITC) [15,36], allicin [37,38], and cinnamaldehyde [36] are electrophiles that covalently modify the channel upon binding through three cysteine residues, Cys621, Cys641, Cys665, or Lys710, located in the cytoplasmic NH_2_-terminal domain in hTRPA1 (Figure 1) [39,40,41,42,43]. However, other studies suggest that some electrophiles and oxidants activate the purified hTRPA1 [19,21] and hTRPA1 expressed in heterologous cells [44] lacking these cysteine residues. Furthermore, the electrophilic compound, *N*-methylmaleimide, binds to cysteine and lysine residues separate from Cys621, Cys641, Cys665 and Lys710 causing conformational changes of purified hTRPA1 [45]. The other group of TRPA1 agonists are non-electrophilic compounds including structurally similar menthol [46], thymol [47], and carvacrol [48]. These bind to the pocket between transmembrane segments S5 and S6, but do not covalently modify cysteine residues [43,49]. The two amino acid residues in segment S5 are crucial for these non-electrophilic agonists’ sensitivity both in human, Ser873 and Thr874, and mouse TRPA1, Ser876 and The877 (Figure 1). TRPA1 is expressed in both unmyelinated and myelinated C- and Aδ-fibers, lungs, skin, bladder, prostate, pancreas, inner ear, and airway epithelial cells. It is both activated and desensitized by Ca^2+^, phospholipase C (PLC) and phosphatidylinositol 4,5-bisphosphate. Alterations in TRPA1 activity are associated with acute and chronic pain, cardiovascular disease, obesity, diabetes, respiratory disease, and digestive disorders [35]. TRPA1 is expressed at a high concentration in trigeminal neurons that innervate nasal and oral epithelium and is co-expressed with TRPV1 [50]. Among the non-neural tissues, TRPA1 is expressed in keratinocytes [51] and in the enteroendocrine cells in the gastrointestinal tract of humans, mice, and rats [52,53]. 

#### 2.1.2. Plant-Derived TRPA1 Agonists

A large number of structurally diverse TRPA1 activators are present in herbs and spices generally consumed with food among many different cultures. Many of the plant-derived molecules have chemesthetic properties and activate both TRPA1 and TRPV1, a structurally related member of the TRP superfamily and which is often co-expressed with TRPA1 in sensory nerve endings with different potencies [50]. Hot peppers, black pepper, garlic, ginger, and sansho in traditional Chinese medicine have a warm or hot nature and contain various TRPV1-activating compounds. While TRPA1- and TRPV1-activating compounds are not identical, some compounds stimulate both TRPV1 and TRPA1. Since the first reports describing AITC, the pungent principal of the genus *Cruciferae* (wasabi) and *Brassica* (mustard) as selective TRPA1 agonist [15], cinnamaldehyde [36], allicin and diallyl disulfide [37,38], the irritant principals of the genus *Cinnamomum* (cinnamon) and *Allium* (garlic) also were identified as TRPA1 activators shortly afterwards (Figure 2). Allicin, diallyl disulfide, and AITC share structural similarities that contain allyl group and sulfur atom that open the TRPA1 channel with different potency. Several derivatives of AITC that contain the isothiocyanate functional group, including benzyl- (in yellow mustard), phenylethyl- (in Brussels sprouts), isopropyl- (in nasturtium seeds), methyl- (in capers) [15,39] and butyl isothiocyanate (in erysimum) [54,55] can also open the TRPA1 channel. However, benzyl thiocyanate, which is isosteric with benzyl isothiocyanate, possesses a thiocyanate functional group of similar size to that of benzyl isothiocyanate, did not activate hTRPA1 [39]. While both compounds possess thiocyanate functional groups of similar size, the single-bond character of the S-C linkage in benzyl thiocyanate reduces its electrophilicity compared with benzyl isothiocyanate (Figure 2). 

Menthol, a naturally occurring cyclic terpene alcohol of plant origin [54], thymol, a monoterpenic phenol compound [47], and carvacrol, an isomeric of thymol [48], the predominant components derived from the thyme plant and oregano essential oils, and eudesmol, an oxygenized sesquiterpene in hop essential oil [55], are representative of non-electrophilic TRPA1 agonists of plant origin. A new class of non-electrophilic TRPA1 agonists have been identified using a structure-based virtual screening approach. While the mechanism of the non-electrophilic agonist has been elucidated recently [16], most commonly known herbs- and spices-derived TRPA1 agonists are electrophiles. 

#### 2.1.3. TRPA1 Activators in Korean Indigenous Plants

Besides the commonly known strong TRPA1 agonists derived from herbs and spices, a variety of activators have also been identified in some wild vegetables. These vegetables are not pungent and, thus, do not cause irritation, and elicit unique flavor that is a distinct sensation from taste or aroma. For example, perillaldehyde and perillaketone, the main flavoring constituents of leaf of *Perilla frutescens*, a plant commonly used in Korean cuisine [56], as well as ligustilide, a major contributor to the aroma of celery (*Apium graveolens* L.) and lovage (*Levisticum officinale* L.) [57], activate TRPA1. To search for potentially unique TRPA1 agonists, 15 Korean indigenous plants were selected that are commonly consumed in Korean cuisine as vegetables (Table 1). The active substances from plants were extracted with 80% ethanol. In cells expressing hTRPA1, Ca^2+^-signaling measurement were used to determine the potency of the extracted compounds in opening hTRPA1 channel. Extract of 13 plants, except Devil’s bush and Kamchatka pleurospermum, showed selective activation of hTRPA1 which was significantly attenuated by both TRPA1 antagonists, ruthenium red and HC-030031 (Figure 3A). The first sprout of Prickly castor oil tree and stem and leaf of Korean mint, in particular, were determined to be active principles for hTRPA1 using commercially available constituents found in the plants. Methyl syringate, a low molecular weight phenolic ester which is a constituent of Prickly castor oil tree selectively activated hTRPA1 but not hTRPV1 [58]. The EC_50_ of methyl syringate was around 70 times larger (507.4 μM) than AITC (7.4 μM), thus, presumed as a weak TRPA1 agonist (Figure 3B). Among the 10 tested constituents of Korean mint, *trans*-*p*-methoxycinnamaldehyde (EC_50_ = 29.8 ± 14.9), methyl eugenol (160.2 ± 21.9), *L*-carveol (189.1 ± 26.8), and *p*-anisaldehyde (546.5 ± 73.0 μM) were found to be selective activators for hTRPA1. Their activation was significantly attenuated by HC-030031 (Figure 3C) [59].

### 2.2. TRPV1

#### 2.2.1. TRPV1 Structure, Expression and Function

Over the course of evolution, TRPV channels have evolved into functionally diverse channels, serving as temperature, chemical and osmotic sensors in vertebrates, chemical and mechanical sensors in nematodes, and hygro- and mechanosensors in insects [13]. TRPV1, the first member of the TRPV subfamily, is a cation non-selective ligand-gated channel expressed in abundances in somatic and visceral sensory neurons in the dorsal root, vagal, and trigeminal ganglia and demonstrates high permeability to Ca^2+^ [23,61]. TRPV1 channel is opened by a large number of exogenous and endogenous mediators, including several vanilloids, high temperature (43–52 °C), acidic stimuli, voltage, and a variety of organic and inorganic molecules. TRPV1 exhibits fourfold symmetry around a central ion pathway formed by transmembrane segments between S5 and S6 and the intervening pore loop, which is flanked by S1–S4 voltage-sensor-like domains [62] (Figure 4). Capsaicin (8-methyl-N-vanillyl-trans-6-nonenamide) is the most prominent TRPV1 agonist of the main pungent ingredient in hot chili peppers that binds to the pocket between cytoplasmic transmembrane segments S3 and S4. The vanillyl moiety of capsaicin binds Tyr511 and Ser512 on S3 via hydrogen bonding while Leu547 and Thr550 on S4 bind the amide group of capsaicin in human TRPV1 [63,64] (Figure 4). Several antagonists have been identified that attenuate TRPV1 function, such as ruthenium red, a noncompetitive antagonist that functions as a pore blocker [65], the synthetic competitive antagonists capsazepine, an analogue of capsaicin [66], and N-(3-methoxyphenyl)-4-chlorocinnamide (SB-366791), which is a potent antagonist for hTRPV1 [67]. TRPV1 is distributed extensively in the unmyelinated C-type sensory nerve fibers and to a lesser extent in myelinated Aδ-type sensory nerve fibers, dorsal root ganglion, trigeminal ganglion, vagal ganglion, central nervous system (CNS), pancreas, liver, lung, heart, gastrointestinal muscle mucosa, gastric mucosal cells and gastric parietal cells, smooth muscle cells, the vascular endothelial cells, the submucosal gland, and inflammatory cells in the respiratory system [68]. 

#### 2.2.2. TRPV1 and Primary Taste Qualities

As reviewed by Smutzer and Devassy [7], oral rinses with capsicum oleoresin (chili extract containing capsaicin and related compounds) decreased perceived intensities of sour taste (citric acid), bitter taste (quinine), and sweet taste (at high sucrose concentrations) without affecting salty taste [69]. In turn, all four primary taste qualities (sweet, sour, salty, and bitter) attenuated the burning sensation of capsaicin in the oral cavity. The decline in the intensity of capsaicin perception following an oral rinse with a taste stimulus occurred most rapidly with citric acid and sucrose. NaCl produced less of an attenuating effect on capsaicin, while the bitter taste stimulus, quinine, had the least effect on capsaicin perception [70]. In another study, all basic tastes aside from umami were influenced by heat, capsaicin, or both [71]. These results suggested an inhibitory effect of oral chemical irritants on the perception of sweet, sour, and bitter taste, and vice versa as well [16,70]. These findings suggest that interactions may occur between trigeminal neurons and taste receptor cells in the oral cavity, or that signal integration between trigeminal neurons and the gustatory system might occur in the central nervous system. However, Cowart [72] simultaneously presented capsaicin and individual primary taste stimuli and reported that the perceived intensity of primary taste qualities was generally unaffected by oral irritation with capsaicin.

Capsaicin is a neurotoxin that affects fibers of the somatosensory lingual nerve surrounding taste buds, but not fibers of the gustatory chorda tympani nerve which synapse with taste receptor cells. Neonate rats made to consume oral capsaicin for 40 days demonstrated a reduction in taste bud volume [73]. Following unilateral transection of the trigeminal lingual nerve, while leaving the gustatory chorda tympani lingual nerve intact, impacts taste receptor cells and alters epithelial morphology of fungiform papillae, particularly during early development. These findings highlight dual roles for the lingual nerve in the maintenance of both gustatory and non-gustatory tissues on the anterior tongue [74].

## 3. Potential Role of TRPA1 and TRPV1 as a Bridge between Taste and Chemesthesis

### 3.1. Cross Talk between TRPA1 and TRPV1

Both TRPV1 and TRPA1 are co-expressed in a large subset of sensory nerves, where they integrate numerous noxious stimuli. Both channels have also been shown to co-express in keratinocytes, mast cells, dendritic cells, and endothelial cells. In cells that co-express TRPA1 and TRPV1, the two TRP channels seem to form a complex at the plasma membrane and influence each other’s function [75,76]. TRPV1-mediated currents in TRPA1-positive neurons characterize small current densities with slow decay, which is caused by TRPA1 channel activities and intracellular Ca^2+^ mobilization. On the other hand, desensitization of TRPV1-mediated current in TRPA1-positive neurons is apparently slow, due to appending TRPA1-mediated current [77]. This cross-communication between TRPA1 and TRPV1 may involve the concentration of cytosolic Ca^2+^, one of the main sensitizers/desensitizers of TRPA1 and TRPV1. These channels can also sensitize each other via a Ca^2+^-dependent signaling pathway [78]. In peripheral sensory fibers, opening TRPA1 and TRPV1 channels stimulates the release of substance P and calcitonin gene-related peptide via axon reflex [79]. Stimulating TRPV1 in uroepithelial keratinocytes causes them to secrete ATP and excite nociceptors [80]. It is suggested that opening TRPA1 channels in keratinocytes may also trigger ATP release and stimulate neighboring somatosensory nerve endings [81,82,83].

### 3.2. TRPA1 and TRPV1 as a Bridge between Taste and Chemesthesis

Taste is a composite of 5 primary taste qualities, salty, sour, sweet, bitter and umami. These taste qualities are detected by a distinct subset of cells within a taste bud. Taste buds are a conglomeration of 50–100 taste receptor cells. Three different subsets of taste receptor cells within the taste bud are characterized as Type I through Type III cells. Among the Type II cells, sweet and umami cells express heterodimeric receptors that are made up of taste receptor type 1 (T1R) G-protein coupled receptors, T1R2/T1R3 and T1R1/T1R3, respectively, while bitter cells express ~25–30 taste receptor type 2 (T2R) G-protein coupled receptors. Downstream of receptors, Type II sweet, bitter and umami taste cells share an intracellular signal transduction mechanism comprising PLCβ2, inositol triphosphate receptor type 3 (IP3), Ca^2+^-dependent monovalent cation channel TRPM5, and voltage-dependent ATP release heterooligomeric channel composed of calcium homeostasis modulator 1 (CALHM1) and 3 (CALHM3). Type III cells express polycystin 2 Like 1 (PKD2L1), TRP cation channel, sour sensing H^+^ channel Otopetrin-1, and carbonic anhydrase 4 [84,85,86]. In fungiform papillae and the soft palate, a subset of Type II cells shows similar but not identical molecular feature with sweet, umami, and bitter taste cells. This newly discovered Type II cell sub-population expresses PLCβ2, IP3, CALHM3, a transcription factor skin head (SKN)-1a, and α subunit of the epithelial Na^+^ channel (ENaC). However, this sub-population of Type II cells lack TRPM5 and guanine nucleotide-binding protein G(t) subunit alpha-3 (GNAT3). SKN-1a-deficient taste buds are predominantly composed of putative non-sensory Type I cells and Type III cells, whereas wild-type taste buds include Type II (i.e., sweet, umami, and bitter taste) cells and Na^+^-taste cells [87]. These subsets of cells with ENaC activity fire action potentials in response to ENaC-mediated Na^+^ influx without changing the intracellular Ca^2+^ concentration and form a channel synapse with afferent neurons involving CALHM1/3 [88].

### 3.3. Chemesthesis and Sour Taste as Co-Mediators of Acid-Evoked Aversion

Sour taste elicits an aversive response to acids. Animals lacking sour taste receptor cells that express PKD2L1 were found to display strong aversion to acids even though the neuronal responses from sour taste receptor cells were completely abolished [89]. Only the animals in which trigeminal TRPV1-expressing neurons were ablated, and that also lacked sour sensing H^+^ channel Otopetrin-1, exhibited a major loss of behavioral aversion to acid. These results suggest that the somatosensory and taste systems serve as co-mediators of acid-evoked aversion [85]. Since TRPA1 channels are also opened by acidic stimuli and are expressed in trigeminal nerve fibers [90], further studies are needed in TRPA1 knockout mice to determine if these channels also contribute to the aversion to sour taste stimuli.

### 3.4. Chemesthesis and Bitter Taste as Co-Mediators of Bitter-Evoked Aversion

Nicotine, present in tobacco plant, is bitter and in behavioral experiments is aversive to animals. Similar to the case with sour taste, the aversive behavior to nicotine involves both somatosensory input and sensing of its bitter taste via the T2Rs/PLCβ2/TRPM5 pathway. TRPM5 pathway is important in detecting most bitter compounds, including quinine and nicotine. In behavioral studies, TRPM5 knockout mice in which trigeminal TRPV1-expressing neurons were ablated still found nicotine aversive, indicating the presence of a TRPM5-independent bitter tasting mechanism. In TRPM5 knockout mice, both behavioral and neural responses were inhibited in the presence of nicotinic acetylcholine receptor (nAChR) blocker, mecamylamine [91]. Thus, with regard to nicotine, its aversive response is due to somatosensory input and its detection by both TRPM5-dependent and a TRPM5-independent pathway that involves nAChRs. A similar mechanism may also be involved in the aversive responses to ethanol at high concentrations [92]. In support of this mechanism, both in rodent and human taste receptor cells, nAChRs are expressed in TRPM5 positive cells [93,94]. These studies suggest that both TRPM5-dependent and TRPM5-independent but nAChR-dependent pathways are present in bitter taste cells. It is important to emphasize that the TRPM5-independent mechanism for detecting the bitter taste of nicotine was revealed after silencing the trigeminal component by injecting capsaicin in neonate rats. Smokers tend to demonstrate higher taste thresholds. This may stem from a reduction in number of fungiform papillae on the tongue [95], suggesting a link between taste and cigrette smoking. Menthol activates TRPM8, and menthol and nicotine are TRPA1 agonists. Menthol decreases oral nicotine aversion in C57BL/6 mice through a TRPM8-dependent mechanism [96]. Allelic variants in TRPA1 might contribute to individual differences in preference for mentholated cigarettes, tendency to smoke cigarettes with higher nicotine levels, extract more nicotine from each cigarette, and/or display greater difficulties in quitting smoking [97,98,99,100]. 

### 3.5. Chemesthesis and Salt Taste

Calcitonin gene-related peptide (CGRP) receptor has been shown to localize in a subset of Type III taste receptor cells, while CGRP was localized in the nerve fibers around the taste buds [89]. A subset of Type III taste receptor cells responded to sub-micromolar concentration of CGRP with an increase in intracellular Ca^2+^ in a PLC-dependent manner. These results raise the possibility that CGRP released from trigeminal nerves can modulate responses in a subset of Type III taste receptor cells that most likely also express the amiloride-insensitive salt receptor [101]. However, at present it is not known if directly opening TRPA1 in nerve fibers around the taste buds will activate CGRP positive Type III cells.

Although, TRPV1 is not expressed in rodent taste receptor cells, TRPV1 agonists capsaicin and resiniferatoxin produced dose-dependent increase in the amiloride-insensitive NaCl chorda tympani taste nerve responses in rodents [102]; at low concentrations enhancing and at high concentrations inhibiting the NaCl chorda tympani responses. TRPV1 knockout mice did not elicit amiloride-insensitive NaCl chorda tympani taste nerve responses. It was hypothesized that in the absence of TRPV1 expression in rodent taste receptor cells, capsaicin and resiniferatoxin most likely produce these effects indirectly via the release of CGRP from trigeminal neurons around the taste bud cells. CGRP then acts on the calcitonin gene-related peptide receptor expressed in Type III cells that presumably also harbor the amiloride-insensitive salt taste receptor(s) [103,104,105]. The effects of capsaicin and resiniferatoxin on amiloride-insensitive NaCl chorda tympani responses are also modulated by *N*-geranyl cyclopropylcarboxamide, an amiloride-insensitive salt taste enhancer that was also found to open both TRPV1 and TRPA1 channels [31,32]. However, at present amiloride-insensitive NaCl chorda tympani responses have not been recorded from TRPA1 knockout mice in the absence and presence of capsaicin, resiniferatoxin, *N*-geranyl cyclopropylcarboxamide and capsiate.

In addition to nerve fibers, TRPV1 is co-expressed with ENaC in kidney cortical-collecting duct cells. Inhibiting TRPV1 activity by high salt in kidney cortical-collecting duct cells enhances α-ENaC expression and opening TRPV1 by capsaicin inhibits the high salt-induced increase in α-ENaC [106]. TRPV1 channel opening prevents high-salt diet-induced hypertension in mice [107]. TRPV1 can be activated when blood pressure increases throughout the body, thereby activating the serine/threonine protein kinase 1 and serum and glucocorticoid-regulated kinase 1 signaling pathways. This results in decreased expression of α-ENaC, which is crucial to the regulation of renal Na^+^ absorption and systemic blood pressure. This effect reduces the reabsorption of Na^+^ and water, ultimately reducing cardiac output and systemic vascular pressure [106]. Although TRPV1 does not seem to be expressed in rodent anterior fungiform taste receptor cells [6], TRPV1 mRNA has been shown to be expressed in stably proliferating human taste cells isolated from fungiform papillae [108]. More recent studies indicate that TRPV1 is also expressed in adult human cultured fungiform taste cells [109]. Cultured human fungiform taste cells in high-salt media increased δ- and γ-ENaC mRNA and protein expression. Capsaicin inhibited the high-salt-induced increase in δ- and γ-ENaC. These effects were TRPV1-dependent. The effects of capsaicin are also mimicked by its non-pungent analogues, capsiate and olvanil. It is interesting to note that non-pungent capsaicin like compounds such as capsiate also open the TRPA1 channel [29].

At present, some studies have implicated a role of TRPV1 in taste perception. However, such studies have not been performed with respect to TRPA1. Since two different TRPA1 knockout mouse models have been generated, these knockout mice models should be used to explore further the relation between chemesthesis and taste and the relationship between TRPV1 and TRPA1. AITC, the powerful plant-derived pungent principal which sensitizes both TRPA1 and TRPV1, significantly suppressed high-sodium responses in chorda tympani taste nerve in mice [110]. This study, again, suggests that chemical that are involved in chemesthesis can modulate primary taste qualities, for example modifying an aversive taste response to high salt. 

### 3.6. Effect of Single-Nucleotide Polymorphisms in TRPA1 and TRPV1 on Chemesthesis and Taste

More recently, it has been suggested that TRP channels provide a link between pain and taste, as they play a role in sensory signaling for taste, thermosensation, mechanosensation, and nociception [8]. Single-nucleotide polymorphisms in TRPA1 and TRPV1 have been shown to affect and modulate both chemesthesis and taste perception. This is illustrated by the observations that genomic variability in the TRPV1 gene correlates with altered capsaicin sensitivity [111,112], alterations in various pain conditions, and with alterations in salty taste sensitivity and salt preference [8,113] in humans. Intronic TRPV1 variants are associated with insensitivity to capsaicin [114], while the coding TRPV1 variant rs8065080 are associated with altered responses to experimentally induced pain [115]. In addition, gain-of-function mutations in TRPV1 are associated with increased pain sensitivity [116]. The effect of TRPV1 modulators on salt taste may involve acute effects on the amiloride-insensitive salt taste responses or long-term effects on ENaC expression in human taste receptor cells. Sweet taste and smell are generally associated with lower pain intensity perception and unpleasantness related with phasic pain compared with bitter taste [117]. 

In addition to salt taste, genetic variation in TRPV1 and TRPA1 also affects other taste qualities. Genetic variation TRPV1 and T2Rs can alter sensations from sampled ethanol and can potentially influence how individuals initially respond to alcoholic beverages [118]. Several variants of the hTRPA1 gene alter the functional properties of the channel in a significant way that are associated with altered pain perception or chemosensory functions in human subjects [43,119,120]. A genetic variant of TRPA1, along with variants in T2R50, and GNAT3, have been linked to personal differences in the taste perception of cilantro [121]. Perception of smell has also been linked to TRPA1 genetic variants. TRPA1 has been shown to be expressed in mouse olfactory bulb [122]. People carrying a TRPA1 SNP (rs11988795) scored better on odor discrimination and had a higher reported H_2_S stimulus intensity. In a Guelph family health study, the rs713598 in the T2R38 bitter taste receptor gene and rs236514 in the *p**otassium inwardly rectifying channel subfamily J member 2 (*KCNJ2*)* sour taste-associated gene showed significant association with phenylthiocarbamide supra threshold sensitivity, and sour taste preference in parents, respectively. In children, rs173135 in KCNJ2 and rs4790522 in the TRPV1 salt taste-associated gene showed significant association with sour and salt taste preferences, respectively [123]. 

The E179K variant of TRPA1 also appears essential for abnormal heat sensation in pain patients. Single amino acid exchange at position 179 in the ankyrin repeat 4 of hTRPA1 seems to be the culprit. TRPA1 wild-type Lys-179 protein when expressed in human embryonic kidney (HEK) cells exhibited normal biochemical properties. It is opened by cold, shows normal trafficking into the plasma membrane and is capable of forming large protein complexes. Its protein expression in HEK cells could be increased at low (4 °C) and high temperatures (49 °C). However, HEK cells expressing the variant Lys179 TRPA1 did not show opening at 4 °C. This can be attributed to its inability to interact with other proteins or other TRPA1 monomers during oligomerization [124]. At present, it not clear how protein genetics of TRPA1 alter the relationship between pain, chemesthesis and taste.

## 4. Conclusions 

Spices and herbs contain chemicals that produce the sensation of chemesthesis by activating thermosensitive TRP channels, including TRPA1 and TRPV1. Recent studies suggest that activation of TRPA1 and/or TRPV1 by spices and herbs involving chemically induced opening of these channels indirectly interacts with the gustatory system. Chemesthetic and taste systems serve as co-mediators of aversive responses to acidic and some bitter taste stimuli. In a mixture with NaCl, TRPV1 agonists induce acute effects on amiloride-insensitive neural responses to NaCl and long-term effects on ENaC expression in salt-sensing human taste receptor cells. These effects most likely involve the release of substance P and CGRP from trigeminal neurons or secondary changes in intracellular calcium that regulate downstream effectors that modulate ENaC expression. It is suggested that application of spicy flavor may be a promising behavioral intervention for reducing high salt intake and lowering blood pressure.

## Figures and Tables

**Figure 1 ijms-22-03360-f001:**
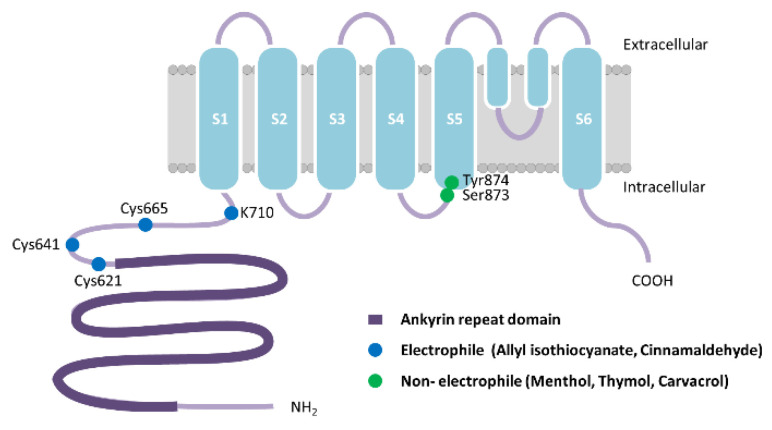
Schematic representation of human transient receptor potential ankyrin 1 (hTRPA1) monomer. Allyl isothiocyanate, allicin, and cinnamaldehyde are electrophiles that covalently modify the channel upon binding through three cysteine residues, Cys621, Cys641, Cys665, or Lys710, located in the cytoplasmic NH_2_-terminal domain in hTRPA1. Menthol, thymol, and carvacrol, structurally similar non-electrophilic agonists, bind to Ser873 and The874 in transmembrane segment S5 in hTRPA1. Adapted from [39].

**Figure 2 ijms-22-03360-f002:**
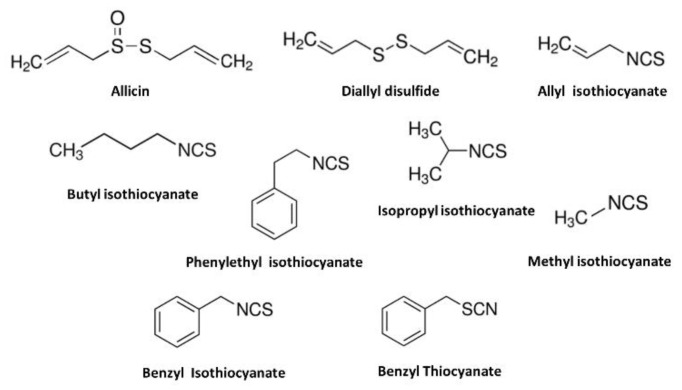
Chemical structures of transient receptor potential ankyrin 1 (TRPA1) agonists that contain allyl group or isothiocyanate functional group found in herb and spices generally consumed with food. Benzyl thiocyanate, an isosteric with benzyl thiocyanate did not activate TRPA1.

**Figure 3 ijms-22-03360-f003:**
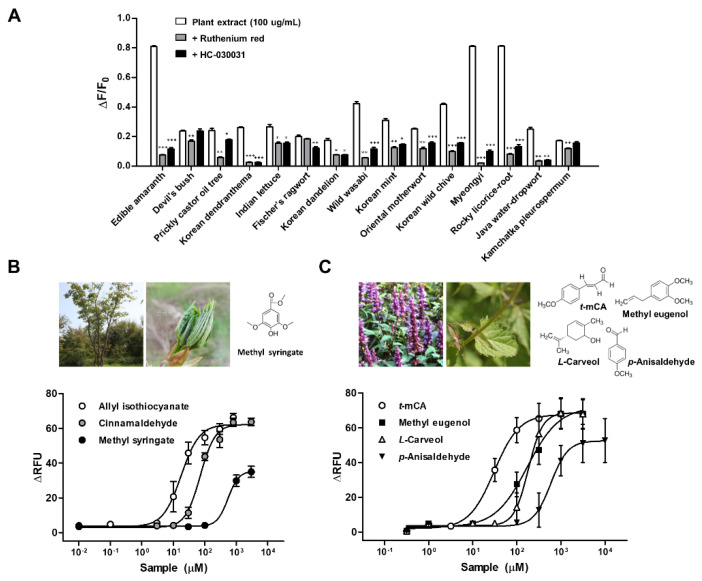
Effect of 15 plant extracts on human transient receptor potential ankyrin 1 (hTRPA1) activity expressed in heterologous cells (**A**). hTRPA1-Flp-In 293^TM^ stable cells were loaded with Fura-2 AM, calcium indicator for ratio-imaging microscopy, and mean changes in fluorescence intensity ratios (F_340_/F_380_) were measured following stimulation with 15 plant extracts (100 μg/mL). The specificity of hTRPA1 stimulation with each plant extract was analyzed using TRPA1 antagonists, ruthenium red and HC-030031. * *p* < 0.05, ** *p* < 0.01, and *** *p* < 0.001 versus the plant extract group using Student’s *t*-test. (**B**,**C**) Concentration–response profiles obtained in hTRPA1-Flp-In 293^TM^ stable cells using Fluo-4, calcium indicator for cell-based calcium signaling assay, from treatment of allyl isothiocyanate, cinnamaldehyde and methyl syringate, a component of prickly castor oil tree (**B**) and *trans-p*-methoxycinnamaldehyde (*t*-mCA), *p*-anisaldehyde, methyl eugenol, *L*-carveol, components of Korean mint (**C**). Data were presented as SEM of at least three separate experiments performed and fitted in GraphPad Prism. ΔRFU = change in Relative Fluorescence Units (y-axis) [59,60].

**Figure 4 ijms-22-03360-f004:**
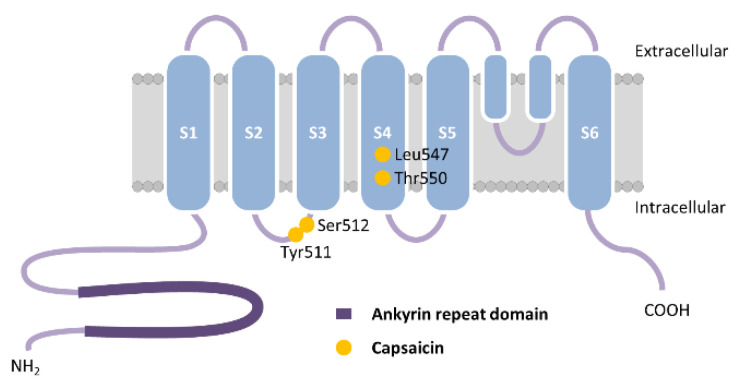
Schematic representation of human transient receptor potential vanilloid 1 (TRPV1) monomer. The vanillyl moiety of capsaicin binds Tyr511 and Ser512 on transmembrane segment S3 via hydrogen bonding while Leu547 and Thr550 on segment S4 bind the amide group of capsaicin in human TRPV1. Adapted from [10].

**Table 1 ijms-22-03360-t001:** List of 15 selective indigenous plants generally consumed in Korean cuisine.

English Name	Local Name	Used Part	Taxon	Scientific Name
Edible amaranth	Bireum	Leaf	Amaranthaceae	*Amaranthus mangostanus* L.
Devil’s bush	Gasiogalpi	Bark	Araliaceae	*Eleutherococcus sessiliflorus* (Rupr. & Maxim.) Maxim.
Prickly castor oil tree	Gaeduleub	Sprout	Araliaceae	*Kalopanax septemlobus* (Thunb.) Koidz.
Korean dendranthema	Sangujeolcho	Whole plant	Compositae	*Dendranthema zawadskii* (Herb) Tzvelev
Indian lettuce	Wanggodeulppaegi	Leaf	Compositae	*Lactuca indica* L.
Fischer’s ragwort	Gomchwi	Leaf	Compositae	*Ligularia fischeri* (Ledeb.) Turcz.
Korean dandelion	Mindeulle	Stem & leaf	Compositae	*Taraxacum platycarpum* Dahlst.
Wild wasabi	Gochunaengi	Leaf	Cruciferae	*Eutrema japonicum (Miq.)* Koidz
Korean mint	Baechohyang	Stem & leaf	Labiatae	*Agastache rugosa* (Fisch. & Mey.) Kuntze
Oriental motherwort	Igmocho	Stem & leaf	Labiatae	*Leonurus japonicus* Houtt.
Korean wild chive	Dallae	Whole plant	Liliaceae	*Allium monanthum* Maxim.
Myeongyi	Ulleungsanmaneul	Leaf	Liliaceae	*Allium ochotense* Prokh.
Rocky licorice-root	Gaehoehyang	Whole plant	Umbelliferae	*Ligusticum tachiroei (Franch. & Sav.)* M.Hiroe & Constance
Java water-dropwort	Dolminari	Stem & leaf	Umbelliferae	*Oenanthe javanica* (Blume) DC.
Kamchatka pleurospermum	Waeusanpul	Stem & leaf	Umbelliferae	*Pleurospermum camtschaticum* Hoffm.

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
