# Peer review of "Interactions between Chemesthesis and Taste: Role of TRPA1 and TRPV1"

_ijms, 2021, doi:10.3390/ijms22073360_

Round 1
Reviewer 1 Report
This is an interesting review about the role of TRPA1 and TRPV1 in the interactions between chemesthesis and taste. The work is novel, well-written and introduces an interesting perspective on the topic. Please find below my suggestions to improve the readability and completeness of the study:
Page 2, line 48.
The authors correctly state that “…it was later revealed that TRPA1 is species-specific in its response to cold temperature”. It would be good and more informative for the reader to provide some more information about the evolutionary significance of TRP channels (TRPA1 and TRPV1 in particular), to better contextualize their role in humans. This is of particular importance, considering that some of the functions played by these channels are conserved across species while other are species-specific (see Himmel and Cox 2020 for a recent review, for example).
Page 3, line 97.
The authors correctly mention that menthol is one of the activators of TRPA1.
It has been shown that, by altering activities of noxious smoke constituents at TRP channels, and covering the bitter taste of nicotine, menthol might alter smoking behaviours. It has been hypothesized, although the results are not congruent, that allelic variants at TRPA1 might contribute to the individual differences in preference for mentholated cigarettes, smoke cigarettes with higher nicotine yields, extract more nicotine from each cigarette, and/or display greater difficulties in quitting smoking (see Willis et al. 2011, Hansen et al. 2017, Uhl et al. 2011, Kozlitina et al. 2019, among others). Considering that the focus of the review is to illustrate the role of TRPA1, TRPV1 and the interactions between chemesthesis and taste, I recommend to include such topic, together with the above references, to give the reader a more comprehensive view on the subject.
Page 5, line 147.
I suggest the authors to add more information regarding TRPA1 expression. For example, highlighting how it is expressed by nociceptors in the lung and airways and in many tissues and how protein genetics variants modulate such expression and functionality (see May et al. 2012 and Meents et al. 2019, for example).
Page 8, line 257.
The authors well-highlight how TRPA1 and TRPV1 may play a potential role as a bridge between taste and chemesthesis. To give the reader a more comprehensive view on the role of such receptors regarding interactions between chemesthesis and taste, I recommend expanding this sub-chapter by discussing how SNPs on TRPA1 and TRPV1 have been shown to affect and modulate both taste perception and chemesthesis. The review, in my opinion, would highly benefit by such inclusion. Some references of such topic include: Allen et al. 2014 (10.1111/acer.12527), Schütz et al. 2014 (10.1371/journal.pone.0095592), Naert et al. 2020 (10.1007/s00424-020-02397-y), Dias et al. 2013 (10.1093/chemse/bjs090), Okamoto et al. 2018 (10.1177/1744806918804439), Chamoun et al. 2018 (10.1080/10408398.2016.1152229), Xu et al. 2007 (10.1152/ajprenal.00347.2007), Aroke et al. 2020 (10.3390/ijms21165929).
Author Response
Responses to Reviewers Comments
We thank reviewer for reading our review very carefully and for your many great suggestions to improve the quality of the review, and the inclusion of many important references that we did not include in the earlier version of the review. Below, please find our point-by-point responses to reviewer’s comments and places in the review where new references and text has been added:
Reviewer #1
In response to comments by Reviewer #1, the changes in the revised version of the review are shown in blue in the red-lined copy.
- Page 2, line 45 (page 2, line 48): The reviewer states that it would be good and more informative for the reader to provide some more information about the evolutionary significance of TRP channels (TRPA1 and TRPV1 in particular), to better contextualize their role in humans. This is of particular importance, considering that some of the functions played by these channels are conserved across species while other are species-specific. “This suggests that while some of TRPA1 attributes, such as, cysteines critical to AITC sensing may be conserved across species, other functions are not [13,17].”Response: Excellent point. Following the suggestion by the reviewer, in the revised version, the following new text has been added in Introduction section: “TRP channels were initially described in Drosophila. Since then TRP channels have been shown to be expressed in animals, choanoflagellates, apusozoans, alveolates, green algae, a variety of eukaryotes, but do not seem to be present in either Archaea or Bacteria. It is suggested that over time the primary role of TRP channels has evolved in signal trans-duction [13].”
- Page 6, line 200 (page 3, line 97): The reviewer states that it has been shown that, by altering activities of noxious smoke constituents at TRP channels, and covering the bitter taste of nicotine, menthol might alter smoking behavior. “3.4. Chemesthesis and bitter taste as co-mediators of bitter-evoked aversion: Smokers tend to demonstrate higher taste thresholds. This may stem from a reduction in number of fungiform papillae on the tongue [88], suggesting a link between taste and cigrette smoking. Menthol activates TRPM8, and menthol and nicotine are TRPA1 agonists. Menthol decreases oral nicotine aversion in C57BL/6 mice through a TRPM8-dependent mechanism [89]. Allelic variants in TRPA1 might contribute to individual differences in preference for mentholated cigarettes, tendency to smoke cigarettes with higher nicotine levels, extract more nicotine from each cigarette, and/or display greater difficulties in quitting smoking [90-93].”Response: Excellent point. Following the suggestion by the reviewer, in the revised version, the following new text and references have been added in the section: “Over the course of evolution, TRPV channels have evolved into functionally diverse channels, serving as temperature, chemical and osmotic sensors in vertebrates, chemical and mechanical sensors in nematodes, and hygro- and mechanosensors in insects [13].”
- Page 11, line 418 (page 5, line 147): The reviewer suggests that the authors should add more information regarding TRPA1 expression. For example, highlighting how it is expressed by nociceptors in the lung and airways and in many tissues and how protein genetics variants modulate such expression and functionality (see May et al. 2012 and Meents et al. 2019, for example). E179K variant of TRPA1 also appear to essential for abnormal heat sensation in pain patients. Single amino acid exchange at position 179 in the ankyrin repeat 4 of hTRPA1 seems to be the culprit. TRPA1 wild type Lys-179 protein when expressed in human embryonic kidney (HEK) cells exhibited normal biochemical properties. It is opened by cold, shows normal trafficking into the plasma membrane and is capable of forming large protein complexes. Its protein expression in HEK cells could be increased at low (4 °C) and high tempertures (49 °C). However, HEK cells expressing the variant Lys179 TRPA1 did not show opening at 4 °C. This can be attributed to its inability to interact with other proteins or other TRPA1 monomers during oligomerization [117]. At present, it not clear how protein genetics of TRPA1 alter the relationship between pain, chemesthesis and taste.” Response: Great suggestion. As recommended by the reviewer, tin he revised version of the review the following new text has been added: “In addition to salt taste, genetic variation in TRPV1 and TRPA1 also affect other taste qualities. Genetic variation TRPV1 and T2Rs can alter sensations from sampled ethanol and can potentially influence how individuals initially respond to alcoholic beverages [111]. Several variants of the hTRPA1 gene alter the functional properties of the channel in a significant way that are associated with altered pain perception or chemosensory functions in human subjects [38,112,113]. A genetic variant of TRPA1, along with variants in T2R50, and GNAT3, have been linked to personal differences in the taste perception of cilantro [114]. Perception of smell has also been linked to TRPA1 genetic variants. TRPA1 has been shown to be expressed in mouse olfactory bulb [115]. People carrying a TRPA1 SNP (rs11988795) scored better on odor discrimination and had a higher reported H2S stimulus intensity. In Guelph family health study, the rs713598 in the T2R38 bitter taste receptor gene and rs236514 in the potassium inwardly rectifying channel subfamily J member 2 (KCNJ2) sour taste-associated gene showed significant association with phenylthiocarbamide supra threshold sensitivity, and sour taste preference in parents, respectively. In children, rs173135 in KCNJ2 and rs4790522 in the TRPV1 salt taste-associated gene showed significant association with sour and salt taste preferences, respectively [116].
- Page 10, line 403 (page 8, line 257): The reviewer states that to give the reader a more comprehensive view on the role of such receptors regarding interactions between chemesthesis and taste, I recommend expanding this sub-chapter by discussing how SNPs on TRPA1 and TRPV1 have been shown to affect and modulate both taste perception and chemesthesis. In addition to salt taste, genetic variation in TRPV1 and TRPA1 also affect other taste qualities. Genetic variation TRPV1 and T2Rs can alter sensations from sampled ethanol and can potentially influence how individuals initially respond to alcoholic beverages [111]. Several variants of the hTRPA1 gene alter the functional properties of the channel in a significant way that are associated with altered pain perception or chemosensory functions in human subjects [38,112,113]. A genetic variant of TRPA1, along with variants in T2R50, and GNAT3, have been linked to personal differences in the taste perception of cilantro [114]. Perception of smell has also been linked to TRPA1 genetic variants. TRPA1 has been shown to be expressed in mouse olfactory bulb [115]. People carrying a TRPA1 SNP (rs11988795) scored better on odor discrimination and had a higher reported H2S stimulus intensity. In Guelph family health study, the rs713598 in the T2R38 bitter taste receptor gene and rs236514 in the potassium inwardly rectifying channel subfamily J member 2 (KCNJ2) sour taste-associated gene showed significant association with phenylthiocarbamide supra threshold sensitivity, and sour taste preference in parents, respectively. In children, rs173135 in KCNJ2 and rs4790522 in the TRPV1 salt taste-associated gene showed significant association with sour and salt taste preferences, respectively [116].”Response: Great point and good topic to add to the review. As suggested by the reviewer the following new text (along with suggested references) has been added: “More recently, it has been suggested that TRP channels provide a link between pain and taste, as they play a role in sensory signaling for taste, thermosensation, mechanosensation, and nociception [8]. Single nucleotide polymorphisms in TRPA1 and TRPV1 have been shown to affect and modulate both chemesthesis and taste perception. This is illustrated by the observations that genomic variability in the TRPV1 gene correlates with altered capsaicin sensitivity [104,105], alterations in various pain conditions, and with alterations in salty taste sensitivity and salt preference [8,106] in humans. Intronic TRPV1 variants are associated with insensitivity to capsaicin [107] while the coding TRPV1 variant rs8065080 are associated with altered responses to experimentally induced pain [108]. In addition, gain-of-function mutations in TRPV1 are associated with increased pain sensitivity [109]. The effect of TRPV1 modulators on salt taste may involve acute effects on the amiloride-insensitive salt taste responses or long-term effects on ENaC expression in human taste receptor cells. Sweet taste and smell are generally associated with lower pain intensity perception and unpleasantness related with phasic pain compared with bitter taste [110].

Reviewer 2 Report
This is an interesting review article on the involvement of the thermosensitive transient receptor potential (TRP) cation channels, particularly the TRPA1 and TRPV1, on taste processing. The following minor changes may improve the quality of this manuscript.
Abstract
The abstract is clear and concise. The authors may, however, explain why they are only addressing the relationship between TRPA1-V1 and some basic tastes, but not others (like sweet). For a non-expert reader, this point is not clear. Besides, in the Introduction section (pag. 69-70), authors refer to “primary taste qualities” (including sweet and umami), and their interactions with these receptors.
Introduction
Line 33: Authors state “Chemosensation refers to the senses of taste and olfaction induced by chemicals”. I am not sure if this is the more precise definition of chemosensation. I would say that this term refers to a biochemical and sensory process of recognition of chemical molecules, by specialized transduction pathways additional to those of smell and taste.
Line 54: “are often co-expressed in trigeminal neurons”. I would say “are often co-expressed in trigeminal ganglion neurons”, to be more precise.
Line 57: “chemically induced activation of these channels”. It would change “activation” for “opening”. What can be activated are the receptors, but channels can be opened or closed.
Line 63: “a synthetic compound, a salt and umami taste enhancer”. Do the authors mean “a synthetic compound, acting as a salt and umami taste enhancer”?
2.1. Plant derived TRPA1 agonists
Lines 87-90: Please, review this sentence “As well as several derivatives of AITC that contain isothiocyanate functional group including benzyl- (BITC; in yellow mustard), phenylethyl- (in Brussels sprouts), isopropyl- (in nasturtium seeds), and methyl- (in capers) [14,31] and butyl-isothiocyanate (in erysimum) [32,33] can be found as activators of TRPA1”, as something seems to be missing to have correct meaning.
Figure 1: What is the meaning of the dotted square surrounding BTC?
2.2. TRPA1 activators in Korean indigenous plants
Line 110: “a variety of activators have also identified in some wild vegetables”. Do the authors mean “a variety of activators have also been identified in some wild vegetables”?
Page 125: “a low molecular weight phenolic ester a constituent of…”. Do the authors mean “a low molecular weight phenolic ester which is a constituent of…”?
Legend Figure 2: “and mean changes in fluorescence intensity ratios (F340/F380) was measured”, should be: “and mean changes in fluorescence intensity ratios (F340/F380) were measured”, as it refers to “changes”. On the other hand, authors should mention what does mean ΛRFU in the Y axis. Why in panel B the measure is mM and in panel C is µM?
3.1. TRPA1 structure, expression and function
To preserve the logical order of descriptions, I would say that section 2 (dedicated to TRPA1 agonists), should go (as further sub-sections) after the sub-section 3.1, since it is also focused on the TRPA1.
Legend of Figure 3: since binding sites for some TRPA1 agonists are represented in this Figure, it would be desirable to add this information in the legend (the same for Figure 4). Besides, in the legend of Figure 4, “monome” should be “monomer”.
3.3. TRPV1 and primary taste qualities
Lines 214-215: “…all four primary taste qualities (sweet, sour, salty, and bitter) attenuated the burning sensation of capsaicin…” What about umami?
Lines 219-220: “These results suggested an inhibitory effect of oral chemical irritants on the perception of sweet, sour, and bitter taste”. And vice versa as well?
3.6. Chemesthesis and sour taste as co-mediators of acid-evoked aversion
Lines 290-292: “Only in animals in which trigeminal TRPV1-expressing neurons were ablated and that also lacked sour sensing H+ channel Otopetrin-1, they exhibited a major loss of behavioral aversion to acid”. Please, rephrase this sentence to make sense… Something like this: “Only the animals in which trigeminal TRPV1-expressing neurons were ablated and that also lacked sour sensing H+ channel Otopetrin-1 exhibited a major loss of behavioral aversion to acid”
3.7. Chemesthesis and bitter taste as co-mediators of bitter-evoked aversion
Line 305: “indicating the presence of a TRPM5-independent bitter tasting mechanism(s).” I would delete (s) because it does not fit with “a (mechanisms)”
Line 306: “In TRPM5 knockout mice both behavioral and neural responses” Responses to what? Do the authors mean “In TRPM5 knockout mice both behavioral and neural aversive responses to bitter taste”?
3.8. Chemesthesis and salt taste
Lines 319-320: The following sentence “In peripheral sensory fibers, activating TRPA1 and TRPV1 channels stimulate the release of substance P and calcitonin gene related peptide (CGRP) via axon reflex” is repeated in the manuscript (lines 250-251). Please, take care of this.
Line 360: “Capsaicin inhibited the HS-induced increase in…” I understand that HS means here high salt. However, since this has not been specified previously, the authors should change it for “high salt”, or, alternatively, provide the abbreviation HS after the first mention of "high salt".
Lines 367-368: “TRPV1 gene correlates with alterations in various pain conditions”. Could the authors provide some examples of such pain conditions? This is an interesting topic worth mentioning.
Line 372: “Bitter taste and unpleasantness may be related to phasic pain compared with.” Compared with what? Something is missing here.
Lines 375-376: “these knockout mice model” Should not be “these knockout mice models”?
Lines 378-379: “significantly suppressed high-sodium responses in chorda tympani taste nerve responses in mice” should be “significantly suppressed high-sodium responses in chorda tympani taste nerve in mice”. By the way, since the term used previously was “high salt” (and HS, I guess), please, consider a homogeneous use of this term, for clarity.
Line 381: “…can modulate primary taste qualities, such as an aversive taste response to high salt” A taste response is not the same that taste quality. Therefore, this sentence should be rewritten to be more precise. Maybe something like this would make more sense: “…can modulate primary taste qualities, for example modifying an aversive taste response to high salt”
Final suggestion
Unless the Journal does not consider it necessary, I would recommend an index of abbreviations and their meanings, given the numerous abbreviations used throughout the manuscript.
Author Response
Responses to Reviewers Comments We thank reviewer for reading our review very carefully and for your many great suggestions to improve the quality of the review, and the inclusion of many important references that we did not include in the earlier version of the review. Below, please find our point-by-point responses to reviewer’s comments and places in the review where new references and text has been added: Reviewer #2 In response to comments by Reviewer #1, the changes in the revised version of the review are shown in red in the red-lined copy. 1. Abstract The reviewer states that the authors may, however, explain why they are only addressing the relationship between TRPA1-V1 and some basic tastes, but not others (like sweet). Response: A good call by the reviewer. The following sentence in the abstract clarifies this point: “TRPA1 and TRPV1 can also indirectly influence some but not all primary taste qualities via the release of substance P and calcitonin gene-related peptide (CGRP) from trigeminal neurons and their subsequent effects on CGRP receptor expressed in Type III taste receptor cells. This point is now also made clear in the Introduction section: “Intriguingly, recent studies suggest that some chemicals that primarily elicit chemesthesis can directly or indirectly influence some but not all primary taste qualities: sweet, bitter, umami, sour, and salty, thus providing a potential linkage between chemesthesis and taste [28].” Following the suggestion by the reviewer, in the revised version, the following new text has been added in abstract: “some, but not all,” 2. Line 34 (line 33): The sentence has been changed to: “Chemosensation refers to a biochemical and sensory process of recognition of chemical molecules, by specialized transduction pathways additional to those of smell and taste.” 3. Line 62 (line 54): The sentence has been changed to: “Among the TRP channels, TRPV1 and TRPA1 are often co-expressed in trigeminal ganglion neurons,…….” 4. Line 65 (line 57): The sentence has been changed to: “These channels can be activated by spices and herbs and are well studied in terms of the molecular mechanisms involving chemically induced opening of these channels.” 5. Line 71 (line 63): The sentence has been changed to: “…..(NGCC), a synthetic compound, acting as a salt and umami taste enhancer [26],………” 6. Line 76. The sentence has been changed to: “Intriguingly, recent studies suggest that some chemicals that primarily elicit chemesthesis can directly or indirectly influence some, but not all, primary………” 7. Figure 1. We have removed the dotted square around benzyl thiocyanate (BTC). The legend has been changed to: “Allyl isothiocyanate, allicin, and cinnamaldehyde are electrophiles that covalently modify the channel upon binding through three cysteine residues, Cys621, Cys641, Cys665, or Lys710, lo-cated in the cytoplasmic NH2-terminal domain in hTRPA1. Menthol, thymol, and carvacrol, structurally similar non-eletrophilic agonists, bind to Ser873 and The874 in transmembrane segment S5 in hTRPA1. Adapted from [34].” 8. 2.1.2. Plant derived TRPA1 agonists. Line 134-139 (lines 87-90): The reviewer asks to please, review this sentence “As well as several derivatives of AITC that contain isothiocyanate functional group including benzyl- (BITC; in yellow mustard), phenylethyl- (in Brussels sprouts), isopropyl- (in nasturtium seeds), and methyl- (in capers) [14,31] and butyl-isothiocyanate (in erysimum) [32,33] can be found as activators of TRPA1”, as something seems to be missing to have correct meaning. Response: In the revised version, the sentence has been changed to: “Several derivatives of AITC that contain the isothiocyanate functional group, including benzyl- (in yellow mustard), phenylethyl- (in Brussels sprouts), isopropyl- (in nasturtium seeds), methyl- (in capers) [15,34] and butyl isothiocyanate (in erysimum) [47,48] can also open TRPA1 channel. While benzyl thiocyanate, which is isosteric with benzyl isothio-cyanate, possesses thiocyanate functional group of similar size to that of benzyl isothio-cyanate, did not activate hTRPA1 [34].” Figure 2. The legend has been changed to: “Chemical structures of transient receptor potential ankyrin 1 (TRPA1) agonists that contain allyl group or isothiocyanate functional group found in herb and spices generally consumed with food. Benzyl thiocyanate, an isosteric with benzyl thiocyanate did not activate TRPA1.” 9. 2.1.3. TRPA1 activators in Korean indigenous plants Line 158 (line 110): We added “been” in the sentence. Line 173 (line 125): The sentence has been changed to: “a low molecular weight phenolic ester which is a constituent………” Figure 3 (Figure 2) legend. “was” has been changed to “were”. The legend ∆RFU is now defined as Relative Fluorescence Units (y-axis). In Figure 2, the panel B axis label has been changed to µM. 10. To preserve the logical order of descriptions, the reviewer suggests that section 2 (dedicated to TRPA1 agonists), should go (as further sub-sections) after the sub-section 3.1, since it is also focused on the TRPA1. Response: Good suggestion. We agree. In the revised version, the section dedicated to TRPA1 agonists has been moved after the section on TRPA1 structure. 11. The legend in Figure 4 has been changed to: “Schematic representation of human transient receptor potential vanilloid 1 (TRPV1) monomer. The vanillyl moiety of capsaicin binds Tyr511 and Ser512 on transmembrane segment S3 via hy-drogen bonding while Leu547 and Thr550 on segment S4 bind the amide group of capsaicin in human TRPV1. Adapted from [10].” 12. 2.2.2. TRPV1 and primary taste qualities Line 242 (lines 214-215): Good point. “In another study, all basic tastes aside from umami were influenced by heat, capsaicin, or both [64].” Lines 244-245 (lines 219-220): The sentence has been changed to: “These results suggested an inhibitory effect of oral chemical irritants on the perception of sweet, sour, and bitter taste, and vice versa as well [16,63].” 13. 3.3. Chemesthesis and sour taste as co-mediators of acid-evoked aversion Lines 310-311 (lines 290-292): The sentence has been changed to: “Only the animals in which trigeminal TRPV1-expressing neurons were ablated and that also lacked sour sensing H+ channel Otopetrin-1, exhibited a major loss of behavioral aversion to acid.” 14. 3.4. Chemesthesis and bitter taste as co-mediators of bitter-evoked aversion Line 326 (line 305): “mechanism(s)” has been changed to “mechanism”. 15. Line 326 (line 306): The sentence has been changed to: “In TRPM5 knockout mice both behavioral and neural responses were inhibited in the presence of nicotinic acetylcholine receptor (nAChR) blocker, mecamylamine [84].” 16. 3.5. Chemesthesis and salt taste Lines 319-320: The following sentence “In peripheral sensory fibers, activating TRPA1 and TRPV1 channels stimulate the release of substance P and calcitonin gene related peptide (CGRP) via axon reflex” is repeated in the manuscript (lines 250-251). Please, take care of this. Response: Thanks. This sentence has been deleted from the text. Line 388 17. Line 388 (line 360): “HS” has been changed to “high salt”. 18. Line 410 (lines 367-368): “TRPV1 gene correlates with alterations in various pain conditions”. The reviewer asks could the authors provide some examples of such pain conditions? This is an interesting topic worth mentioning. Response: Thanks. Good suggestion. The following new text and references have been added: “Intronic TRPV1 variants are associated with insensitivity to capsaicin [107] while the coding TRPV1 variant rs8065080 are associated with altered responses to experimentally induced pain [108]. In addition, gain-of-function mutations in TRPV1 are associated with increased pain sensitivity [109]. The effect of TRPV1 modulators on salt taste may involve acute effects on the amiloride-insensitive salt taste responses or long-term effects on ENaC expression in human taste receptor cells.” 19. Line 415 (line 372). Thanks. The sentence has been changed to: “Sweet taste and smell are generally associated with lower pain intensity perception and unpleasantness related with phasic pain compared with bitter taste”. 20. Line 394 (lines 375-376): Thanks. Good suggestion. The sentence has been changed to: “…… these knockout mice models should be used to explore further the relation between chemesthesis and taste.…..”. 21. Line 397 (lines 378-379): Thanks. Good suggestion. The sentence has been changed to: AITC, the powerful plant-derived pungent principal which sensitizes both TRPA1 and TRPV1, significantly suppressed high-sodium responses in chorda tympani taste nerve in mice. 22. Line 398 (line 381): Thanks. Good suggestion. The sentence has been changed to: “.….. can modulate primary taste qualities, for example modifying an aversive taste response to high salt. 23. Abbreviations: We have been added “an index of abbreviations and their meanings”.
